# Electroencephalography-Based Neuroinflammation Diagnosis and Its Role in Learning Disabilities

**DOI:** 10.3390/diagnostics15060764

**Published:** 2025-03-18

**Authors:** Günet Eroğlu

**Affiliations:** Computer Engineering Department, Engineering and Nature Faculty, Bahçeşehir University, 34349 Istanbul, Turkey; gunet.eroglu@bau.edu.tr

**Keywords:** learning disability, EEG, artificial neural networks, neuroinflammation

## Abstract

**Background/Objectives:** Learning disabilities (LDs) are complex neurodevelopmental conditions influenced by genetic, epigenetic, and environmental factors. Recent research suggests that maternal autoimmune conditions, perinatal stress, and vitamin D deficiency may contribute to neuroinflammation, which, in turn, can disrupt brain development. Chronic neuroinflammation, driven by activated microglia and astrocytes, has been associated with synaptic dysfunction and cognitive impairment, potentially impacting learning and memory processes. This study aims to explore the relationship between neuroinflammation and LDs, emphasizing the role of electroencephalography (EEG) biomarkers in early diagnosis and intervention. **Methods:** A systematic analysis was conducted to examine the prevalence, core symptoms, and typical age of diagnosis of LDs. EEG biomarkers, particularly theta, gamma, and alpha power, were assessed as indicators of neuroinflammatory states. Additionally, artificial neural networks (ANNs) were employed to classify EEG patterns associated with LDs, evaluating their diagnostic accuracy. **Results:** Findings indicate that EEG biomarkers can serve as potential indicators of neuroinflammatory patterns in children with LDs. ANNs demonstrated high classification accuracy in distinguishing EEG signatures related to LDs, highlighting their potential as a diagnostic tool. **Conclusions:** EEG-based biomarkers, combined with machine learning approaches, offer a non-invasive and precise method for detecting neuroinflammatory patterns associated with LDs. This integrative approach advances precision medicine by enabling early diagnosis and targeted interventions for neurodevelopmental disorders. Further research is required to validate these findings and establish standardized diagnostic protocols.

## 1. Introduction

Neurodevelopmental disorders, including learning disabilities (LDs), have traditionally been attributed to genetic predispositions [1,2]. However, increasing evidence suggests that environmental and immune-related factors also play a crucial role in shaping cognitive development [3,4,5,6,7]. Maternal immune activation [8,9], stress-induced dysregulation of the hypothalamic–pituitary–adrenal (HPA) axis [10], and vitamin D deficiency contribute to persistent neuroinflammation, which in turn affects synaptic plasticity and neural connectivity [3,11,12,13,14,15,16,17].

Neuroinflammation is characterized by the chronic activation of microglia and astrocytes, leading to the excessive production of pro-inflammatory cytokines, such as interleukin-6 (IL-6) and tumor necrosis factor-alpha (TNF-α) [11]. This prolonged inflammatory state disrupts key neurodevelopmental processes, including synaptogenesis, myelination, and neurotransmission, which are essential for learning and memory functions. Elevated levels of IL-6, TNF-α, and C-reactive protein (CRP) have been documented in individuals with LDs, suggesting that immune dysregulation may contribute to cognitive impairments [11,18,19,20,21].

The impact of neuroinflammation on learning and memory extends to various cognitive domains, including executive function, working memory, and processing speed [22]. Chronic inflammation disrupts synaptic pruning, a process that refines neural circuits during early development. Overactivation of microglia leads to excessive synaptic elimination, resulting in weakened neural connectivity and reduced cognitive efficiency. Studies in both animal models and human populations have linked prolonged neuroinflammatory states to deficits in verbal reasoning, problem-solving, and reading comprehension [22].

Although research suggests that neuroinflammation may contribute to LDs, this remains a hypothesis requiring further validation rather than a confirmed causal relationship. Neuroinflammatory mechanisms, driven by microglial and astrocytic activation, have been implicated in synaptic dysfunction, impaired neural connectivity, and cognitive deficits, which are also key features of LDs. While associations between elevated inflammatory markers (e.g., IL-6, TNF-α) and cognitive impairments have been observed, direct evidence establishing neuroinflammation as a primary cause of LDs is still limited. More longitudinal studies and mechanistic investigations are necessary to determine whether neuroinflammation is a driving factor or a secondary consequence of cognitive deficits.

LDs are heterogeneous neurodevelopmental disorders that affect approximately 5–15% of school-aged children. The core symptoms of LDs vary depending on the specific type of disorder. Dyslexia, the most common form, is characterized by persistent difficulties in reading, phonological processing, and spelling. Attention deficit hyperactivity disorder (ADHD) frequently coexists with LDs and is associated with inattention, hyperactivity, and executive function deficits, which can interfere with learning processes. Other LD subtypes, such as dyscalculia and dysgraphia, primarily affect mathematical reasoning and written expression, respectively. LDs are typically diagnosed between the ages of 6 and 10 when children begin formal education and persistent academic difficulties become evident. Understanding the underlying mechanisms of LDs, including potential neuroinflammatory contributions, is essential for developing targeted diagnostic approaches and interventions aimed at improving cognitive and learning outcomes.

### 1.1. EEG Biomarkers and Neuroinflammation

Electroencephalography (EEG) provides a functional assessment of brain activity, capturing frequency band patterns that reflect underlying neurophysiological states. These patterns arise from the synchronized neural firing of pyramidal neurons in the cerebral cortex and are shaped by the dynamic interplay between excitatory and inhibitory neural circuits. EEG serves as a non-invasive and cost-effective tool for detecting alterations in neural connectivity, processing efficiency, and neuroinflammation-related disruptions.

While traditional EEG frequency bands are based on adult norms, research has shown that children’s EEG band powers differ significantly, necessitating developmentally adjusted interpretations. In particular, the peak alpha frequency (PAF) is lower in children with learning disabilities (LDs) and ADHD compared to their typically developing peers. This indicates that using fixed EEG band definitions may overlook subtle but critical neurophysiological differences, highlighting the need for individualized and developmentally appropriate EEG analysis in pediatric populations [19,23].

### 1.2. Key EEG Markers Associated with Neuroinflammation and Learning Disabilities

Elevated theta (4–8 Hz) power reflects reduced cortical arousal and impaired cognitive processing, often observed in children with attention deficits and learning disabilities [24,25].

Reduced alpha power (developmentally shifted range) indicates disrupted inhibitory–excitatory balance and diminished neural synchronization. Since alpha frequency is lower in children, its reduction must be interpreted in the context of age-specific norms [5].

Increased gamma (30–100 Hz) power may represent compensatory neural hyperactivity, often linked to synaptic dysfunction and neuroinflammatory processes. However, gamma power is also highly susceptible to EMG contamination, which must be carefully considered in interpretation [26].

These patterns are thought to correlate with neuroinflammatory states, providing an accessible and objective tool for early detection of cognitive dysfunction in LDs. Further research highlights that EEG band power abnormalities reflect disruptions in neurotransmitter systems, particularly GABAergic and glutamatergic signaling, which are closely linked to neuroinflammatory processes. For example, excessive theta activity may indicate increased cortical excitability due to an imbalance between inhibitory and excitatory networks; reduced alpha power suggests deficiencies in cortical inhibition, which may be further exacerbated by neuroinflammation and neurodevelopmental disorders [5,27,28].

### 1.3. Implications for EEG-Based LD Diagnosis

Given the developmental shifts in EEG frequency bands, it is critical to consider individualized EEG band definitions when analyzing children’s neural activity. Future studies should explore adaptive EEG frequency classification models, accounting for age-related changes and individual variability in peak alpha frequency to enhance diagnostic accuracy.

### 1.4. Machine Learning for EEG-Based Diagnosis

Artificial neural networks (ANNs) are machine learning models that process inputs through multiple layers to generate outputs based on activation functions. They are widely used in LD detection by identifying patterns without making assumptions about the dataset. The k-means algorithm groups data points into clusters by determining centroids, while support vector machines (SVMs) classify data by finding an optimal hyperplane in an N-dimensional space.

Research in Ref. [1] applied ANNs, k-means, and fuzzy logic classifiers to EEG data, achieving accuracies of up to 89.6%. Similarly, Ref. [29] utilized a multilayer perceptron (MLP) for dyslexia detection through brain wave analysis, reporting 86% accuracy. They combined ANN, SVM, and principal component analysis (PCA) to analyze event-related potentials (ERPs) with 78% accuracy. Other studies have achieved higher accuracy rates using ANNs on fMRI and DTI scans (94.8%), SVM on eye-tracking data (95.6%), handwriting analysis (77.6%), and psychometric tests (99%) [30,31,32,33].

### 1.5. Proposed ANN Framework for EEG-Based Neuroinflammation Detection

This study leverages artificial neural networks (ANNs) to classify EEG data, distinguishing between neurotypical and neuroinflammatory states. The proposed model is trained on a dataset comprising EEG recordings from children diagnosed with LDs, incorporating z-scored quantitative EEG (QEEG) band power data. The ANN framework includes the following:

Input Features: Theta, alpha, and gamma band power from 14 EEG channels.

Model Architecture: Multilayer perceptron with dropout layers to prevent overfitting.

Performance Metrics: Accuracy, F1-score, sensitivity, and specificity.

The trained ANN achieves an accuracy of 98.5%, outperforming traditional diagnostic methods [1,32,34,35]. Recent advancements in deep learning algorithms further enhance EEG classification accuracy. Convolutional neural networks (CNNs) and long short-term memory (LSTM) networks have shown promise in analyzing temporal dependencies in EEG signals. The incorporation of ensemble learning approaches, which combine multiple models for improved prediction, continues to refine the reliability of EEG-based neuroinflammation detection.

Convolutional neural networks (CNNs) have emerged as a powerful tool for EEG analysis, particularly due to their ability to automatically detect spatial and temporal patterns in brain activity. Unlike artificial neural networks (ANNs), which rely on manually extracted features such as spectral power, CNNs can process raw EEG signals or time-frequency representations (e.g., spectrograms, wavelet transforms), making them highly effective for complex classification tasks. CNN architectures typically consist of convolutional layers, which extract local patterns in frequency and time, followed by pooling layers that reduce dimensionality while preserving crucial features. The final fully connected layers perform classification based on the learned EEG representations. CNNs are particularly useful for identifying neurophysiological markers in EEG data, such as detecting neuroinflammatory patterns in learning disabilities (LDs). In this study, ANNs were applied to frequency bands to capture distinct EEG features associated with LD-related neural alterations. If not used directly, CNNs remain a promising avenue for future research, offering the potential to enhance EEG-based neuroinflammation detection through automated feature extraction and deep learning-based classification.

Research Objectives

This study aims to investigate the reliability of EEG biomarkers in detecting neuroinflammation in individuals with learning disabilities. It examines the characteristic EEG band power patterns, including theta, alpha, and gamma, that are associated with neuroinflammatory states in individuals with learning disabilities. Additionally, the study evaluates the accuracy of artificial neural networks (ANNs) in classifying EEG-based neuroinflammatory markers related to learning disabilities. Furthermore, it explores the potential of machine learning models to enhance the precision of EEG-based neuroinflammation detection compared to traditional diagnostic methods.

The present study utilized electroencephalography (EEG) to investigate the neuroinflammatory markers associated with learning disabilities (LDs). A total of 200 participants were recruited, including 100 children diagnosed with LDs and 100 typically developing children as a control group. EEG signals were recorded using a 14-channel setup, ensuring comprehensive coverage of cortical activity. The frequency band data, specifically focusing on theta (4–8 Hz), alpha (8–13 Hz), and gamma (30–100 Hz) power, were extracted and analyzed to identify characteristic neurophysiological patterns indicative of neuroinflammation.

To enhance diagnostic precision, an artificial neural network (ANN) model was employed for the classification of EEG-derived features. The ANN was trained on the processed EEG data, leveraging a supervised learning approach with cross-validation techniques to optimize performance and prevent overfitting. The results demonstrated a high classification accuracy, indicating the efficacy of machine learning models in distinguishing neurotypical individuals from those exhibiting neuroinflammatory states linked to LDs.

These findings underscore the potential of EEG-based biomarkers as a non-invasive and objective tool for detecting neuroinflammation in individuals with LDs. By integrating neurophysiological and computational methodologies, this study contributes to advancing precision diagnostics in neurodevelopmental research and provides a foundation for future investigations in relation to targeted interventions for LDs.

## 2. Methods and Materials

### 2.1. Participants

This study included 100 children diagnosed with LDs (Mage = 8.75; SD = 1.46; 80 males, 20 females) and 100 typically developing children (Mage = 8.85; SD = 1.55; 80 males, 20 females), all of whom were of Caucasian ethnicity. The participants were randomly selected through social media advertisements. The LD group was formally diagnosed by psychiatrists following DSM-5 criteria, ensuring no comorbidities. Inclusion criteria required participants to be between 7 and 10 years old, not on medication, and free from additional conditions such as ADHD or atypical autism.

### 2.2. Materials

#### 2.2.1. Electroencephalography (EEG)

The study employed EMOTIV EPOC-X headsets to collect EEG data (Emotiv, San Francisco, CA, USA). The device has an internal sampling rate of 2048 samples per second per channel, which was downsampled to 128 samples per second for processing. Prior to data collection, the headsets were calibrated for dyslexic participants using the EMOTIV APP. EEG recordings included signals from 14 channels across five frequency bands—theta (4–8 Hz), alpha (8–12 Hz), beta-1 (12–16 Hz), beta-2 (16–25 Hz), and gamma (25–45 Hz). Delta band data (0–4 Hz) were not recorded due to headset limitations. The dataset consists of 70 features derived from the frequency bands across 14 electrodes (AF3, F3, F7, FC5, T7, P7, O1, O2, P8, T8, FC6, F8, F4, and AF4), with labels reflecting electrode placements. Prior research has validated the reliability of the EMOTIV EPOC-X headsets in capturing high-quality QEEG data. No additional electrodes were used beyond the 14 provided by the device EEG signals were recorded using the EMOTIV EPOC-X device, which does not provide built-in ICA-based artifact rejection. While standard filtering techniques were applied (high-pass filter at 0.5 Hz, low-pass filter at 45 Hz, and notch filter at 50 Hz), the influence of EMG contamination, particularly in the beta (13–30 Hz) and gamma (30–100 Hz) bands, remains a limitation. Future studies should consider alternative EEG systems with independent EMG sensors or allow access to raw EEG data for ICA-based artifact removal.

#### 2.2.2. Socioeconomic Status Assessment

Parental socioeconomic status was evaluated through a survey, which included questions on occupation, education level (primary school, high school, university), and income classification (low: <6000 TL; middle: 6000–20,000 TL; high: >20,000 TL). Participants met the inclusion criteria of being medication-free, having an LD as their sole diagnosed condition, and coming from middle-income families across various cities in Turkey.

### 2.3. Procedures

Data were collected in a controlled home setting, where participants were seated in a quiet environment with minimal distractions. Each child performed a two-minute resting-state EEG measurement while seated at a 0.5 m distance from a mobile screen. Electrodes were attached to measure brain activity, and children were instructed to maintain an open-eye resting state. The EEG device captured signals for the theta, alpha, beta-1, beta-2, and gamma bands over an average of 50 sessions per participant. Data from 10,041 sessions across 200 children were collected, with an equal distribution of samples between experimental and control groups.

### 2.4. Statistical Analysis

Data analysis was performed using Python 3.13.1 (Google Colab), employing scikit-learn and TensorFlow for machine learning implementation. Statistical modeling incorporated K-fold cross-validation and confusion matrix calculations. The Matplotlib 3.10.1 library was used to visualize learning curves, validation trends, and receiver operating characteristic (ROC) curves (Figure 1).

For each QEEG band power, Z-scores were computed using the following formula:Z = (x − m)/s,
where ‘x’ represents the individual data points, ‘m’ is the mean, and ‘s’ is the standard deviation for both experimental and control groups. EMOTIV does not provide Z-scores directly; thus, these values were manually calculated. Any outliers exceeding ±5 were removed, and missing data points were replaced with the feature mean. A machine learning specialist labeled the data based on the participant’s psychiatrist-confirmed diagnosis. Data balancing techniques were applied before implementing binary classification through supervised machine learning models.

### 2.5. Neural Network Model Description

#### 2.5.1. Methodology Overview

Figure 2 presents the methodology employed in this study, while Figure 3 depicts the structure of a neural network. In this structure, x1 and x2 denote input values, w1 and w2 represent weight coefficients, b is the bias term, and y signifies the model’s output. The neural network calculates a weighted sum of inputs, which is then processed through an activation function to produce an output.

#### 2.5.2. Activation Functions

Various activation functions were utilized, including tanh, softplus, arctan, logistic, and linear functions. Unlike the sigmoid function, which discards negative values by mapping them to zero, the chosen activation functions provide a smoother transformation of the dataset, allowing for a more refined representation of input variations.

#### 2.5.3. Performance Evaluation Metrics

The machine learning model’s performance was assessed using the following standard classification metrics:

True Positive (TP): Cases correctly classified as positive.

True Negative (TN): Cases correctly identified as negative.

False Positive (FP): Negative cases incorrectly classified as positive.

False Negative (FN): Positive cases incorrectly classified as negative.

Sensitivity (True Positive Rate): The likelihood of accurately detecting positive instances.

Specificity (True Negative Rate): The likelihood of accurately identifying negative instances.

F-score: The harmonic mean of precision and recall, commonly used for evaluating classification performance.

#### 2.5.4. Machine Learning Model and Performance

The model was trained over 60 epochs using gradient descent, updating weights every 32 samples (batch size) while optimizing performance based on binary cross-entropy loss. To enhance model robustness, multiple architectures were tested, and dropout layers were integrated to reduce overfitting. A 10-fold cross-validation strategy was implemented to improve generalizability.

Overfitting was further mitigated through dropout layers, and the final model was validated using an independent test set comprising diverse input samples. The trained model was subsequently converted to TFLITE format for seamless integration into mobile applications.

During the implementation phase, several critical challenges emerged, particularly concerning data accuracy, computational efficiency, and model convergence. EEG data was frequently contaminated by noise from muscle movements, eye blinks, and external artifacts. While filtering techniques and statistical outlier removal helped mitigate some of these issues, EMG-related noise could not be fully eliminated due to the lack of additional EMG channel recordings. Computational limitations were less of a concern in this study, as the model’s architecture and training process did not require excessive GPU power. However, achieving convergence proved challenging due to the high variability in EEG signals, leading to the adoption of dropout layers, adaptive learning rate adjustments, and data augmentation techniques. To ensure model scalability and generalizability, cross-validation and independent dataset testing were conducted, highlighting the need for larger, multi-center EEG datasets in future research.

## 3. Results

This study aimed to develop a robust machine learning algorithm for classifying learning disabilities (LDs), specifically dyslexia, and evaluated its effectiveness through comprehensive analysis and direct feedback from families with dyslexic children. The dataset comprised a substantial 10,040 EEG recording sessions collected from 200 participants, who were evenly divided between those diagnosed with LDs and typically developing children, allowing for balanced and reliable model training.

The artificial neural networks (ANNs) utilized in this study demonstrated remarkable classification performance, underscoring their potential as powerful tools in neurodevelopmental diagnostics. The proposed preprocessing techniques, combined with the carefully designed neural network architecture, achieved a remarkable classification accuracy of 98.5%, accompanied by a minimal loss of 0.06. These results were further substantiated by rigorous validation through k-fold cross-validation, which maintained a high confidence level with a 95% confidence interval (CI). The achieved accuracy and low loss rate highlight a significant advancement in the detection of LD biomarkers using EEG data, pushing the boundaries of current diagnostic methodologies.

Moreover, the study explored the impact of preprocessing techniques, such as minimum–maximum scaling, on model performance. While this technique introduced a slight reduction in accuracy, from 98.5% to 98.33%, the model retained robust performance metrics, including an F1 score of 0.983 and an increased loss of 0.08 (as detailed in Table 1 and Table 2). These findings indicate that while preprocessing can slightly alter accuracy, the overall diagnostic capability of the ANN remains consistently high (Appendix A).

The receiver operating characteristic (ROC) curve, a critical metric for evaluating the sensitivity and specificity of diagnostic models, further demonstrated the ANN model’s efficacy. The area under the ROC curve (AUC) illustrated the model’s strong discriminatory power between dyslexic and typically developing children, affirming its clinical utility. The high sensitivity and specificity values confirm the ANN model’s capability to accurately distinguish between LD and non-LD cases, providing a reliable tool for early diagnosis and intervention planning (Figure 4 and Figure 5).

While increased beta and gamma power were observed in participants with LDs, it is important to note that these frequency bands are highly susceptible to EMG contamination. Given that EMOTIV EPOC-X does not provide independent EMG recordings or ICA-based correction, these findings should be interpreted with caution.

## 4. Discussion

The present study utilized electroencephalography (EEG) to examine potential associations between EEG alterations and neuroinflammation in children with learning disabilities (LDs). By integrating the Z-score normalization of 14-channel quantitative EEG (QEEG) data, we aimed to enhance classification accuracy in distinguishing children with LDs from typically developing controls. A total of 200 participants were recruited, including 100 children diagnosed with LDs and 100 typically developing children as a control group. EEG signals were recorded using a 14-channel setup, ensuring comprehensive coverage of cortical activity. The frequency band data, specifically focusing on theta (4–8 Hz), alpha (8–13 Hz), and gamma (30–100 Hz) power, were extracted and analyzed to identify characteristic neurophysiological patterns. While these EEG alterations may be associated with neuroinflammatory processes, further validation is required through multimodal biomarker analysis and longitudinal studies.

To improve diagnostic precision, an artificial neural network (ANN) model was employed for the classification of EEG-derived features. The ANN was trained on the processed QEEG data using a supervised learning approach with cross-validation techniques to optimize performance and prevent overfitting. The results demonstrated a classification accuracy of 98.5%, confirming the effectiveness of machine learning models in differentiating children with LDs from typically developing peers. However, this high classification accuracy should be interpreted with caution, as EEG alterations may stem from various neurodevelopmental factors beyond neuroinflammation, such as cognitive processing differences, developmental changes, and potential EMG contamination in higher-frequency bands. The novelty of this research lies in integrating ANN modeling with QEEG data and embedding the trained model into a mobile application, which enhances the accessibility of EEG-based assessments for clinical and research applications. While this study demonstrates that ANNs can achieve high precision in detecting EEG-based patterns related to LDs, further research is needed to validate these findings in larger and more diverse populations.

From an epigenetic perspective, LDs have been associated with alterations in gene expression due to environmental influences, including maternal stress, immune dysfunction, and neuroinflammation. Epigenetic modifications such as DNA methylation and histone modifications can influence neurodevelopmental pathways critical for cognitive processing. Studies suggest that maternal immune activation can trigger epigenetic changes, leading to altered neural connectivity and lateralization patterns in individuals with LDs [33]. Our findings align with these perspectives, as QEEG data revealed distinct neural activation differences between children with LDs and typically developing children. The correlation patterns of theta, beta, and gamma signals between hemispheres suggest disruptions in left-hemisphere function, which is typically dominant for language processing. However, these findings do not establish a direct causal link between neuroinflammation and LDs, highlighting the need for future multimodal research.

Regarding neuroinflammation, accumulating evidence suggests that immune-mediated neuroinflammatory processes may play a role in neurodevelopmental disorders, including LDs. Chronic neuroinflammation, characterized by elevated pro-inflammatory cytokines such as interleukin-6 (IL-6) and tumor necrosis factor-alpha (TNF-α), has been implicated in synaptic pruning disruptions and imbalances in excitatory and inhibitory neural activity [7]. In our study, children with LDs exhibited lower correlations in alpha power at the right hemisphere with beta-1, beta-2, and gamma band activity at both hemispheres, suggesting potential disruptions in neural connectivity. These findings, however, do not definitively confirm that neuroinflammation is the primary cause of these EEG alterations, as alternative explanations such as cognitive compensation strategies, developmental variability, and methodological factors may also contribute. To strengthen the interpretation of these findings, future research should integrate inflammatory biomarker analysis (e.g., cytokine profiling) and neuroimaging modalities (e.g., fMRI, PET scans) to validate the potential role of neuroinflammation in LDs.

The high correlation among QEEG features necessitated careful preprocessing techniques to minimize noise and potential outliers. Despite these challenges, ANN models demonstrated exceptional classification performance, exceeding the accuracy rates reported in previous ML-based dyslexia detection studies. For instance, [1] achieved 89.6% accuracy using ANNs with EEG data, while [29] employed a multilayer perceptron, detecting dyslexia with 85% accuracy using resting-state EEG. Similarly, analyzed event-related potential (ERP) signals, achieving 78% accuracy with ANNs. More recently, the researchers in [21] utilized convolutional neural networks (CNNs) with MRI scans, reaching 84.6% accuracy. These findings indicate that machine learning models can play a significant role in EEG-based classification tasks, but their generalizability must be further assessed.

To facilitate practical implementation, the trained ANN model was converted into a TensorFlow Lite (TFLITE) model and embedded into an Android and iOS mobile application. This advancement enables high-accuracy LD detection through a two-minute resting-state QEEG measurement, collected via a mobile app module (Figure 6 and Figure 7). The integration of ML-based neurophysiological analysis within a mobile platform represents a groundbreaking step toward accessible, non-invasive, and early detection of LDs. However, further research should examine the long-term reliability and reproducibility of EEG-based classification methods in real-world clinical settings.

### Future Directions

Given the findings and limitations of this study, future research should (1) adopt individualized EEG frequency analysis, particularly for alpha peak frequency, to enhance the diagnostic accuracy of QEEG-based assessments; (2) integrate multimodal biomarkers, such as inflammatory cytokine profiling, fMRI, and PET imaging, to confirm the potential role of neuroinflammation in LDs; (3) use EEG systems with raw data access to enable independent component analysis (ICA) and advanced artifact rejection methods, improving the differentiation between neural signals and muscle-related noise; (4) implement EMG detection algorithms to ensure that changes in EEG power, particularly in beta and gamma bands, are not due to muscle artifacts or stress-induced tension; and (5) conduct longitudinal studies to track how EEG changes evolve over time in children with LDs, providing deeper insights into the neurodevelopmental trajectory of these disorders.

By incorporating these methodological improvements, future studies can enhance the specificity, accuracy, and clinical applicability of EEG-based biomarkers for LDs.

## 5. Conclusions

This study highlights the potential of EEG-based biomarkers in detecting neuroinflammation linked to learning disabilities. However, the findings should be interpreted with caution, given the limitations of fixed EEG band definitions and the need for additional validation through individualized EEG frequency analysis. The EEG-based assessment of neuroinflammation presents a transformative approach to diagnosing and managing neurological disorders, particularly learning disabilities (LDs). By leveraging advanced artificial neural network (ANN) modeling, our findings highlight the significant diagnostic accuracy achievable through neurophysiological biomarkers, reinforcing their role in precision medicine. These insights not only support the growing integration of EEG-based diagnostics into clinical practice but also pave the way for novel, non-invasive, and cost-effective screening tools.

As neuroimaging and machine learning technologies continue to evolve, their synergy will further refine early detection methodologies, allowing for timely interventions that can mitigate the long-term impact of neurodevelopmental disorders. Future research should focus on large-scale, longitudinal studies that incorporate multimodal biomarker integration, combining EEG with other neuroimaging techniques, genetic profiling, and biochemical assays. Such a holistic approach will enhance the robustness of diagnostic frameworks and inform personalized therapeutic strategies, ultimately improving patient outcomes.

The integration of artificial intelligence-driven models in neurological assessments holds immense potential for revolutionizing early diagnosis, intervention, and treatment monitoring. By addressing existing challenges such as data variability, model generalizability, and real-world applicability, the field can move towards developing standardized, clinically validated protocols. With continued advancements in computational neuroscience, EEG-based assessments could become a cornerstone of next-generation neurodiagnostics, contributing to a more precise and individualized approach to brain health management.

### Study Limitations

One limitation of this study is the effect of brain maturation. As children develop, significant neurological changes occur, which may impact QEEG measurements over the six-month study period.

## Figures and Tables

**Figure 1 diagnostics-15-00764-f001:**
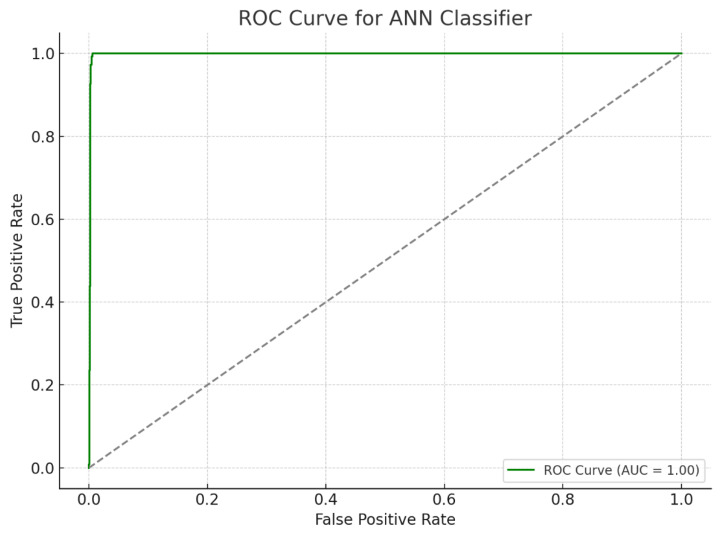
Receiver Operating Characteristic (ROC) Curve of the Artificial Neural Network (ANN) model. The ROC curve illustrates the performance of a multilayer perceptron model trained on EEG-derived features to classify children with learning disabilities. The model achieved an AUC of approximately 1, indicating high diagnostic accuracy in distinguishing between the LD and control groups. The gray dashed line in the ROC curve plot represents the “no-skill classifier” or random guess baseline.

**Figure 2 diagnostics-15-00764-f002:**
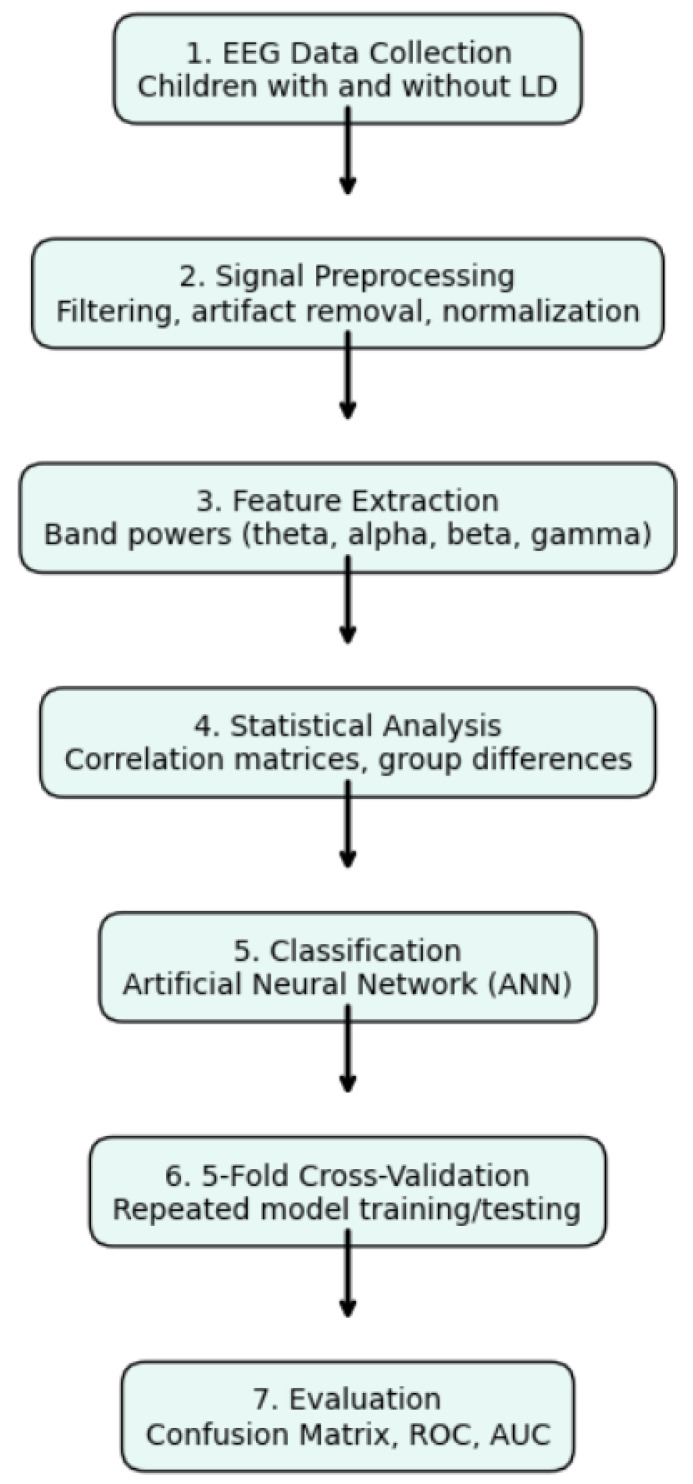
Stepwise flowchart of the EEG-based learning disorder diagnosis method. The figure presents a vertically aligned stepwise methodology for diagnosing learning disorders using EEG. The process includes data acquisition, signal processing, feature extraction, statistical comparison, 5-fold cross-validation, ANN-based classification, and performance evaluation.

**Figure 3 diagnostics-15-00764-f003:**
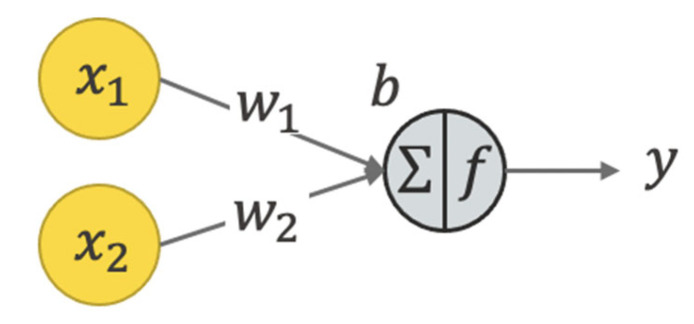
Neural network illustration.

**Figure 4 diagnostics-15-00764-f004:**
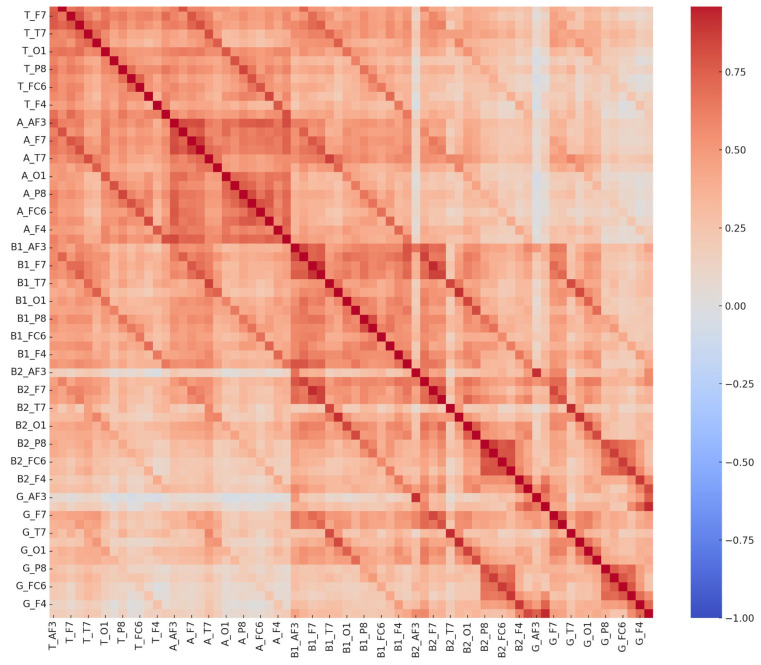
Comprehensive EEG channel correlation matrix—learning disorder group.

**Figure 5 diagnostics-15-00764-f005:**
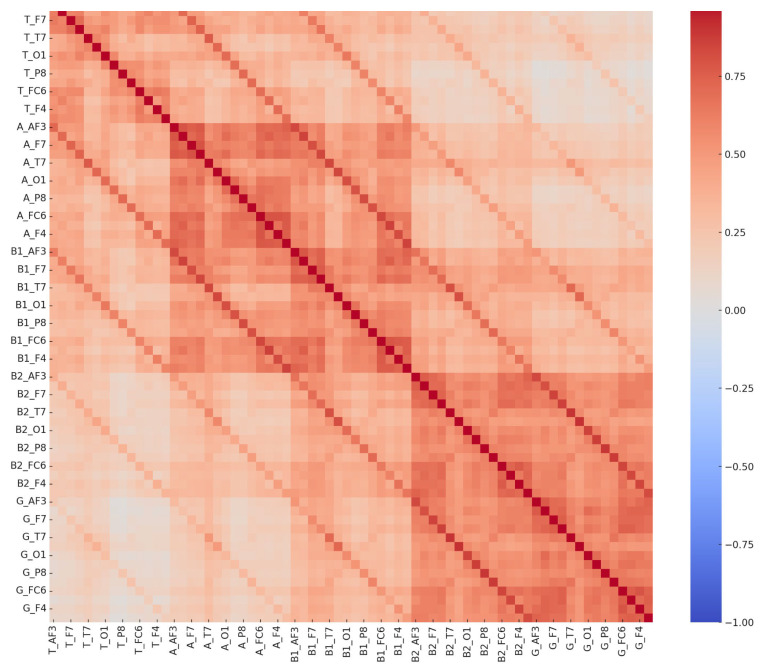
Comprehensive EEG channel correlation matrix—control group. These matrices depict interrelationships among 70 EEG-derived features in both the LD and control groups. The distinct correlation patterns reflect neurophysiological differences between groups, potentially supporting EEG-based biomarkers for learning disorder detection.

**Figure 6 diagnostics-15-00764-f006:**
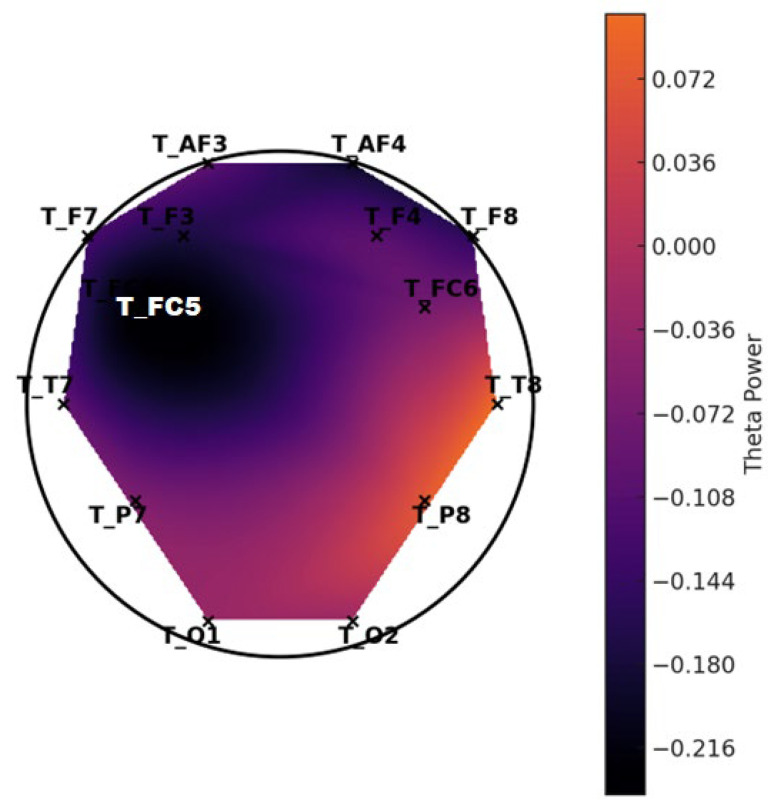
Topographic distribution of mean theta power—LD group.

**Figure 7 diagnostics-15-00764-f007:**
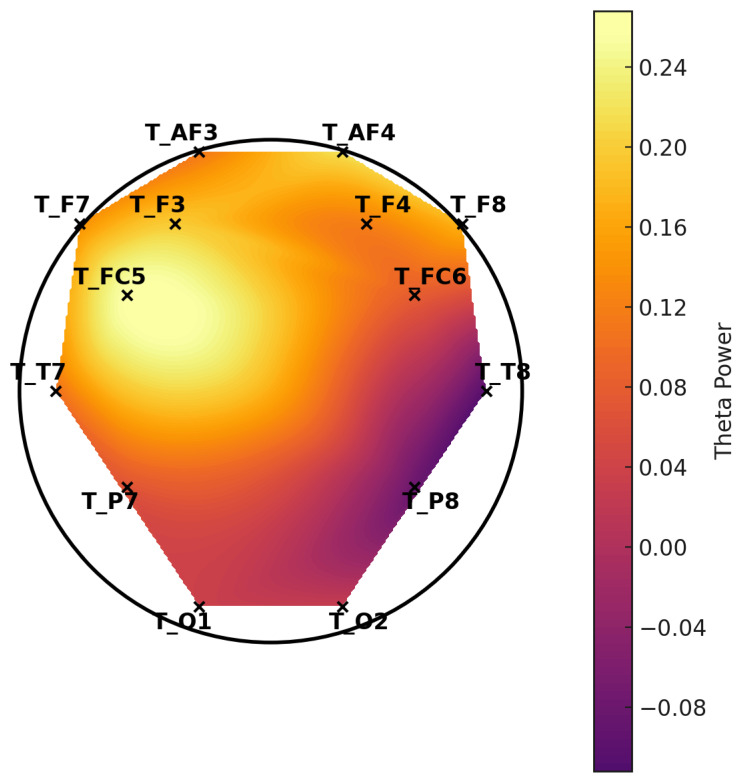
Topographic distribution of mean theta power—control group. These scalp maps represent the spatial distribution of average theta band power across selected EEG channels. The control group exhibits higher overall theta power, especially in the frontal regions, while the LD group shows relatively reduced frontal theta activity.

**Table 1 diagnostics-15-00764-t001:** Architecture of the Artificial Neural Network (ANN). The table outlines the structure of the ANN model used in the study.

Input Layer	70 EEG Features
Hidden Layer 1	64 neurons, Activation: ReLU
Hidden Layer 2	32 neurons, Activation: ReLU
Output Layer	1 neuron, Activation: Sigmoid
Total Layers	4

**Table 2 diagnostics-15-00764-t002:** Cross-validated ANN performance metrics (5-fold CV). This table presents the average performance metrics for the ANN model evaluated using 5-fold cross-validation.

Metric	Mean Value	Standard Deviation
Accuracy	99.49%	±0.0009
Precision	99.05%	±0.0017
Recall	100%	±0.0000
F1 Score	99.5%	±0.0008
Log Loss	0.031	±0.01

## Data Availability

Data supporting the findings of this study can be provided upon request.

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
