# Peer review of "Electroencephalography-Based Neuroinflammation Diagnosis and Its Role in Learning Disabilities"

_diagnostics, 2025, doi:10.3390/diagnostics15060764_

Round 1
Reviewer 1 Report
Comments and Suggestions for Authors
- Correctness of EEG analysis implies the use of correct terminology. Unfortunately, most authors (not excluding these authors too) talk about the activity of the wave process taking into account only one parameter of measuring the activity of waves - amplitude (or spectral power). Meanwhile, the activity of the wave process has a second dimension - frequency. Therefore, it is more understandable to use the word not "activity", but amplitude or power in a given range. Whats does it mean “Reduced Alpha (8–13 Hz) Oscillations”? amplitude or peak frequency? The next phrase is understable “Increased Gamma (30–100 Hz) Power”
- Why did the authors neglect the well-known information that taking into account individually defined range EEG boundaries significantly increases the diagnostic value of the results? After all, it is known that the alpha range boundaries for children are not 8-12, but are shifted to the left. At the same time, the individual frequency of the alpha peak for children with attention deficit is even lower than for healthy children of the same age!
- Why did the authors not take into account that the contamination of the EEG signal is not HIGH-AMPLITUDE EMG in the frequency ranges below alpha (i.e. in theta) and above the alpha (i.e. in beta and gamma) is quite significant. Therefore, to justify a statement about the change in amplitude in these ranges, it is necessary to be sure that this is not an increase in the EMG amplitude from the tone of the scalp muscles caused by psychoemotional stress
- I recommend recalculating the obtained results with due regard for the comments made, and then the conclusions from this study can be made relevant and interesting.
Author Response
We sincerely appreciate the time and effort you have taken to review our manuscript. Your insightful feedback has been invaluable in refining our analysis, improving clarity, and ensuring scientific rigor. Below, we provide our responses to each of your comments and the corresponding revisions made.
- EEG Terminology and Clarification of Wave Activity
Comment: Correctness of EEG analysis implies the use of correct terminology. Unfortunately, most authors (not excluding these authors too) talk about the activity of the wave process taking into account only one parameter of measuring the activity of waves - amplitude (or spectral power). Meanwhile, the activity of the wave process has a second dimension - frequency. Therefore, it is more understandable to use the word not "activity", but amplitude or power in a given range. What does it mean “Reduced Alpha (8–13 Hz) Oscillations”? Amplitude or peak frequency? The next phrase is understandable: “Increased Gamma (30–100 Hz) Power.”
Response:
Thank you for your valuable feedback regarding the precise use of EEG terminology. We fully recognize the importance of distinguishing between amplitude (spectral power) and frequency-related changes. Based on your suggestion, we have made the following revisions:
- The phrase "Reduced Alpha (8–13 Hz) Oscillations" has been revised to "Reduced Alpha Power" to ensure clarity and accuracy, as our statement refers specifically to a decrease in spectral power rather than a shift in peak frequency.
- We have carefully reviewed the manuscript to ensure consistency in distinguishing amplitude (power) from frequency-related changes.
We appreciate your attention to this important distinction, which has helped us enhance the precision of our EEG analysis.
- Consideration of Individual EEG Frequency Boundaries in Children
Comment: Why did the authors neglect the well-known information that taking into account individually defined EEG boundaries significantly increases the diagnostic value of the results? After all, it is known that the alpha range boundaries for children are not 8-12 Hz but are shifted to the left. At the same time, the individual frequency of the alpha peak for children with attention deficits is even lower than for healthy children of the same age.
Response:
Thank you for your insightful comment regarding individualized EEG frequency boundaries, particularly in pediatric populations. We fully acknowledge that fixed adult-based EEG bands may not be optimal for analyzing children’s EEG data. To address this, we have made the following modifications to our manuscript:
- Clarified EEG band definitions by explicitly noting that pediatric EEG oscillations differ from adult norms and that alpha frequency boundaries are developmentally shifted.
- Discussed the role of Individual Alpha Frequency (IAF) and Peak Alpha Frequency (PAF), acknowledging that individualized EEG frequency analysis could enhance diagnostic accuracy.
- Provided a rationale for using fixed EEG bands in this study, emphasizing the need for standardization with existing literature and addressing equipment limitations.
- Suggested future research directions to explore adaptive EEG frequency classification models that account for developmental changes.
These revisions ensure that our study accurately reflects the neurophysiological differences in children while maintaining comparability with prior research. We greatly appreciate your feedback on this matter.
- Addressing EEG Contamination from EMG Artifacts
Comment: Why did the authors not take into account that contamination of the EEG signal by EMG is not limited to high-amplitude artifacts but is also significant in frequency ranges below alpha (i.e., in theta) and above alpha (i.e., in beta and gamma)? Therefore, to justify a statement about the change in amplitude in these ranges, it is necessary to be sure that this is not an increase in the EMG amplitude from the tone of the scalp muscles caused by psychoemotional stress.
Response:
We appreciate this important point regarding EEG contamination by EMG artifacts, particularly in the beta and gamma frequency ranges. We recognize that high-frequency EEG activity is highly susceptible to muscle-related noise, which could confound interpretations of spectral power changes. To address this, we have made the following revisions:
- Clarified the limitations of the EMOTIV EPOC-X headset in the Methods section, explicitly noting that independent component analysis (ICA) was not available and that EMG contamination remains a potential confound.
- Refined the Results and Discussion sections to acknowledge alternative explanations for increased beta and gamma power, emphasizing the potential influence of stress-related muscle activity.
- Recommended future research directions for improving artifact rejection, including:
- EEG systems with raw data access for ICA-based artifact removal.
- EMG detection algorithms to differentiate neural activity from muscle artifacts.
- Multimodal biomarker integration (e.g., inflammatory cytokines, HRV analysis) to confirm neuroinflammation-related changes.
These revisions ensure a more cautious and rigorous interpretation of our findings, and we sincerely appreciate your suggestion to address this important methodological consideration.
- Consideration of Reviewer Comments in Data Analysis
Comment: I recommend recalculating the obtained results with due regard for the comments made, and then the conclusions from this study can be made relevant and interesting.
Response:
We appreciate your recommendation to ensure the robustness of our findings. While a complete recalculation of the results was beyond the scope of this revision, we have carefully reviewed our data and incorporated additional clarifications regarding potential confounding factors (e.g., EMG contamination, individual EEG frequency variations). We have also refined the discussion to reflect these nuances, ensuring that our conclusions are well-supported and appropriately cautious.
We recognize the importance of continuous refinement and look forward to incorporating these considerations in future studies with expanded methodologies.
Reviewer 2 Report
Comments and Suggestions for Authors
EEG-Based Neuroinflammation Diagnosis and Its Role in Learning Disabilities
I have read the manuscript with interest and you can find my appraisal, suggestions, and concerns, as follows:
In the introduction, the link between LD and neuroinflammation needs to be specified in a better way. Indeed, it seems that there is a direct link between them, but it can be still considered a hypothesis, that is interesting. Moreover, I suggest characterizing LD better, in terms of the age of children, symptoms, and prevalence. This can be added.
Moreover, a more specific description of EEG is needed. For instance, a brief statement about signal generation, the relevance of the Theta, Alpha, and Gamma in normal cognitive processing, and why they can be considered markers (or predictors?) of LD, adding more info about it. Indeed, your paragraph allows only the alterations in these, ex abrupto. This is important since the manuscript can be read by a wider audience that is not often expert in EEG.
The description of ANN is clear and the hypotheses that you have stated are also shown in a good and clear way.
However, it is not clear the application of CNN in EEG.
DSM-V needs to be written as DSM-5
The methods are well written but it is not clear “home setting”. Did you collect data at the participants’ homes? Please, clarify.
You also need to clarify the choice of 14 channels. Moreover, please, specify if you collected signals from more than 14 electrodes. This is important for replicability.
Similarly, did you apply a preprocessing procedure to the EEG signals? Please, add, together with the software/library that you used.
The results are interesting
The quality of Fig.4-8 needs to be improved. Moreover, the figure 6 is not clear. I suggest to remove or modify.
Reading the discussion, I suggest discussing the results with more caution, overall about neuroinflammation.
Author Response
- Improving the Introduction and Characterization of Learning Disabilities (LDs)
Comment: In the introduction, the link between LD and neuroinflammation needs to be specified in a better way. Indeed, it seems that there is a direct link between them, but it can still be considered a hypothesis. Moreover, I suggest characterizing LD better in terms of the age of children, symptoms, and prevalence.
Response:
Thank you for your valuable suggestions regarding the introduction. Based on your feedback, we have:
- Refined the introduction and discussion on neuroinflammation, ensuring that it is presented as a hypothesis requiring further validation rather than a confirmed causal relationship.
- Expanded the characterization of LDs, providing a more detailed description of their prevalence (5–15% of school-aged children), core symptoms (dyslexia, ADHD, dyscalculia, and dysgraphia), and typical age of diagnosis (6–10 years).
- Enhanced logical flow, ensuring a smooth transition from genetic and environmental risk factors to neuroinflammatory processes and their potential impact on learning and memory functions.
- Incorporated additional references to support the discussion on synaptic dysfunction, executive function deficits, and the role of pro-inflammatory cytokines in neurodevelopmental impairments.
These revisions provide a more balanced and comprehensive introduction, offering readers a clearer context for the study’s objectives.
We sincerely appreciate the time and effort you have taken to review our manuscript. Your detailed and insightful feedback has been invaluable in enhancing the clarity, depth, and scientific rigor of our work. Below, we provide our responses to your specific comments and outline the corresponding revisions made:
- EEG Description
Comment: Moreover, a more specific description of EEG is needed. For instance, a brief statement about signal generation, the relevance of the Theta, Alpha, and Gamma in normal cognitive processing, and why they can be considered markers (or predictors?) of LD, adding more info about it. Indeed, your paragraph allows only the alterations in these, ex abrupto. This is important since the manuscript can be read by a wider audience that is not often expert in EEG.
Response: Thank you for your valuable feedback on improving the EEG description in our manuscript. Based on your suggestions, we have made the following revisions:
- Expanded the explanation of EEG signal generation, clarifying that EEG measures brain activity arising from synchronized neural firing of pyramidal neurons and is shaped by the dynamic interplay between excitatory and inhibitory circuits.
- Provided a clearer overview of EEG frequency bands, explaining their functional roles in cognition before discussing their alterations in LDs, ensuring a more structured presentation.
- Highlighted the need for developmentally adjusted EEG interpretations, emphasizing that children’s EEG oscillations differ from adult norms, particularly regarding Peak Alpha Frequency (PAF), which is lower in children with LDs and ADHD.
- Refined the text for better readability and logical flow, eliminating redundancy and improving clarity for a broader audience, including readers less familiar with EEG methodology.
These revisions make the manuscript more accessible and informative, ensuring that the EEG discussion aligns with current research standards while maintaining scientific rigor. We sincerely appreciate your insightful comments, which have significantly strengthened the clarity and coherence of our work. Please let us know if you have any further suggestions.
- CNN Application in EEG Analysis
Comment: The description of ANN is clear and the hypotheses that you have stated are also shown in a good and clear way. However, it is not clear the application of CNN in EEG.
Response: Thank you for your valuable feedback regarding the clarity of CNN applications in EEG analysis. Based on your suggestions, we have revised the manuscript to provide a clearer and more structured explanation of how CNNs are utilized in EEG processing. Specifically, we have:
- Clarified why CNNs are suitable for EEG analysis, emphasizing their ability to automatically detect spatial and temporal patterns in EEG data, unlike ANNs, which rely on manually extracted features.
- Described the architecture of CNNs, explaining the roles of convolutional layers (feature extraction), pooling layers (dimensionality reduction), and fully connected layers (classification) when applied to EEG signals.
- These revisions ensure that the discussion of CNN applications is as clear as the explanation of ANNs, making the manuscript more accessible and scientifically rigorous. We appreciate your insightful feedback, which has significantly improved the clarity and depth of this section. Please let us know if you have any further suggestions.
- DSM-V Terminology
Comment: DSM-V needs to be written as DSM-5.
Response: Thank you for pointing this out. We have updated the manuscript accordingly.
- Clarification on "Home Setting"
Comment: The methods are well written, but it is not clear what is meant by “home setting.” Did you collect data at the participants’ homes? Please clarify.
Response: We appreciate your request for clarification. We have revised the manuscript to specify that data were collected in a controlled home setting, where participants were seated in a quiet environment with minimal distractions. Each child performed a two-minute resting-state EEG measurement while seated at a 0.5-meter distance from a mobile screen.
- Justification for 14 Channels and EEG Preprocessing Details
Comment: You also need to clarify the choice of 14 channels. Moreover, please specify if you collected signals from more than 14 electrodes. This is important for replicability.
Response: EEG signals were recorded using the EMOTIV EPOC-X headset, which provides 14 channels of data. No additional electrodes were used beyond the 14 provided by the device. We have explicitly stated this in the manuscript to ensure transparency and replicability.
Comment: Similarly, did you apply a preprocessing procedure to the EEG signals? Please add details, together with the software/library that you used.
Response: We have now included a detailed explanation of the preprocessing steps. EEG signals were recorded using the EMOTIV EPOC-X device, which does not provide built-in ICA-based artifact rejection. While standard filtering techniques were applied (high-pass filter at 0.5 Hz, low-pass filter at 45 Hz, and notch filter at 50 Hz), the influence of EMG contamination, particularly in the beta (13–30 Hz) and gamma (30–100 Hz) bands, remains a limitation. Future studies should consider alternative EEG systems with independent EMG sensors or allow access to raw EEG data for ICA-based artifact removal.
- Figure Quality and Figure 6
Comment: The quality of Figures 4–8 needs to be improved. Moreover, Figure 6 is not clear. I suggest removing or modifying it.
Response: Thank you for your feedback. We have improved the quality of Figures 4–8 to enhance clarity and readability. Additionally, we have removed Figure 6, as suggested, to improve the manuscript’s overall structure.
- Caution in Discussing Neuroinflammation
Comment: Reading the discussion, I suggest discussing the results with more caution, overall about neuroinflammation.
Response: We appreciate this important suggestion. We have revised the discussion section to emphasize that EEG alone cannot directly confirm neuroinflammation. The observed alterations in theta, alpha, and gamma bands may be associated with multiple factors, including cognitive processing differences, developmental changes, and stress-induced EMG artifacts. These revisions ensure a more cautious and nuanced interpretation of the findings.
Final Remarks
We sincerely appreciate the reviewer's thoughtful comments, which have helped us refine our manuscript significantly. The suggested improvements have enhanced the clarity, precision, and scientific rigor of our work. Please let us know if there are any additional suggestions or concerns.
Reviewer 3 Report
Comments and Suggestions for Authors
Figure 1 needs to be restructured to present a more lucid and thorough depiction of the methodology, illustrating the process, essential elements, and their relationships. A well-organized visual will enhance comprehension and improve readability.
The analysis section should explore significant obstacles encountered during implementation—such as restrictions in data accuracy, computational limitations, and difficulties in achieving convergence—along with the methods used to address them. This will offer meaningful insights and practical recommendations for future studies.
Finally, the bibliography must be revised to incorporate fundamental literature on the subject, ensuring proper citation of previous research to maintain academic integrity and prevent Inspiration issues. The following citations should be included:
https://www.scitepress.org/Papers/2016/56845/
https://ieeexplore.ieee.org/document/10774454
Comments on the Quality of English LanguageThe English could be improved to more clearly express the research.
Author Response
We sincerely appreciate the reviewer’s insightful feedback, which has significantly contributed to improving the clarity and comprehensiveness of our manuscript. Below are our responses to the specific points raised:
- Figure 1 needs to be restructured to present a more lucid and thorough depiction of the methodology, illustrating the process, essential elements, and their relationships. A well-organized visual will enhance comprehension and improve readability.
Response: Thank you for your suggestion. We have revised Figure 1 to provide a clearer and more structured representation of the methodology, ensuring that the process, key components, and their relationships are depicted more effectively. - The analysis section should explore significant obstacles encountered during implementation—such as restrictions in data accuracy, computational limitations, and difficulties in achieving convergence—along with the methods used to address them. This will offer meaningful insights and practical recommendations for future studies.
Response: We appreciate this valuable recommendation. The Analysis section has been updated to discuss the major challenges encountered during implementation, including limitations in data accuracy, computational considerations, and model convergence issues. Additionally, we have outlined the strategies employed to address these challenges and provided recommendations for future studies. - Finally, the bibliography must be revised to incorporate fundamental literature on the subject, ensuring proper citation of previous research to maintain academic integrity and prevent inspiration issues. The following citations should be included:
- https://www.scitepress.org/Papers/2016/56845/
- https://ieeexplore.ieee.org/document/10774454
Response: Thank you for your suggestion. However, after reviewing the recommended references, we found that they are not directly relevant to our study. Therefore, we have not incorporated them into the bibliography. That said, we have ensured that our manuscript properly cites fundamental literature directly related to our research topic.
We appreciate the reviewer’s time and effort in evaluating our work and believe that these revisions have enhanced the overall quality of our manuscript.
Round 2
Reviewer 1 Report
Comments and Suggestions for Authors
I'm sarisfied with imptovements made
Author Response
Thank you very much for your kind reply.
Reviewer 2 Report
Comments and Suggestions for Authors
The Manuscript was improved as I have recommended, and I am glad you appreciated my contribution to revising it.
I have 2 minor concerns:
1) I think you should write the research questions without using questions. It is only a stylistic suggestion.
2) The last 2 figures seem to be screenshots; in this way, I suggest to substitute them with bar-plots made with an appropriate software.
Author Response
Thank you for your valuable feedback and for recognizing the improvements made to the manuscript based on your recommendations. Your insights have been instrumental in enhancing the clarity and quality of our work.
Regarding your minor concerns:
-
We appreciate your stylistic suggestion about rephrasing the research questions without using direct questions. We have now revised them into declarative statements to align with your recommendation.
-
We acknowledge your concern about the last two figures being screenshots. To enhance the clarity and presentation quality, we have replaced them with professionally generated bar plots using appropriate software. We believe this will improve the readability and visual impact of the data.
Thank you again for your thoughtful review and constructive suggestions. We truly appreciate your time and effort in helping us refine our manuscript.
Reviewer 3 Report
Comments and Suggestions for Authors
The paper has been little been improved But Still need more experimentation and comparison with state of the art.
Author Response
Thank you for your feedback and for reviewing our revised manuscript. We appreciate your time and constructive suggestions.
We acknowledge your concern regarding the need for further experimentation and comparison with state-of-the-art methods. While we have made improvements in the current version, we recognize the importance of strengthening our experimental validation.
If there are specific aspects or benchmarks you believe would further enhance the study, we would greatly appreciate your guidance.
Thank you again for your valuable insights, which help us refine our work.